# Development of Reduced Peptide Bond Pseudopeptide Michael Acceptors for the Treatment of Human African Trypanosomiasis

**DOI:** 10.3390/molecules27123765

**Published:** 2022-06-11

**Authors:** Santo Previti, Roberta Ettari, Carla Di Chio, Rahul Ravichandran, Marta Bogacz, Ute A. Hellmich, Tanja Schirmeister, Sandro Cosconati, Maria Zappalà

**Affiliations:** 1Department of Chemical, Biological, Pharmaceutical and Environmental Sciences, University of Messina, Viale Stagno d’Alcontres 31, 98166 Messina, Italy; rettari@unime.it (R.E.); cdichio@unime.it (C.D.C.); 2Department of Clinical and Experimental Medicine, University of Messina, Via C. Valeria, 98125 Messina, Italy; 3DiSTABiF, University of Campania Luigi Vanvitelli, Via Vivaldi 43, 81100 Caserta, Italy; rahul.ravichandran@unicampania.it (R.R.); sandro.cosconati@unicampania.it (S.C.); 4Institute of Organic Chemistry & Macromolecular Chemistry, Friedrich-Schiller-University of Jena, Humboldtstraße 10, 07743 Jena, Germany; marta.bogacz@uni-jena.de (M.B.); ute.hellmich@uni-jena.de (U.A.H.); 5Centre for Biomolecular Magnetic Resonance (BMRZ), Goethe University Frankfurt, Max von Laue Str. 9, 60438 Frankfurt, Germany; 6Institute of Pharmaceutical and Biomedical Sciences, University of Mainz, Staudingerweg 5, 55128 Mainz, Germany; schirmei@uni-mainz.de

**Keywords:** sleeping sickness, Michael acceptors, peptide backbone modifications, antitrypanosomal agents, rhodesain, pseudopeptides

## Abstract

Human African Trypanosomiasis (HAT) is an endemic protozoan disease widespread in the sub-Saharan region that is caused by *T. b. gambiense* and *T. b. rhodesiense*. The development of molecules targeting rhodesain, the main cysteine protease of *T. b. rhodesiense*, has led to a panel of inhibitors endowed with micro/sub-micromolar activity towards the protozoa. However, whilst impressive binding affinity against rhodesain has been observed, the limited selectivity towards the target still remains a hard challenge for the development of antitrypanosomal agents. In this paper, we report the synthesis, biological evaluation, as well as docking studies of a series of reduced peptide bond pseudopeptide Michael acceptors (**SPR10**–**SPR19**) as potential anti-HAT agents. The new molecules show *K*_i_ values in the low-micro/sub-micromolar range against rhodesain, coupled with *k*_2nd_ values between 1314 and 6950 M^−1^ min^−1^. With a few exceptions, an appreciable selectivity over human cathepsin L was observed. In in vitro assays against *T. b. brucei* cultures, **SPR16** and **SPR18** exhibited single-digit micromolar activity against the protozoa, comparable to those reported for very potent rhodesain inhibitors, while no significant cytotoxicity up to 70 µM towards mammalian cells was observed. The discrepancy between rhodesain inhibition and the antitrypanosomal effect could suggest additional mechanisms of action. The biological characterization of peptide inhibitor **SPR34** highlights the essential role played by the reduced bond for the antitrypanosomal effect. Overall, this series of molecules could represent the starting point for further investigations of reduced peptide bond-containing analogs as potential anti-HAT agents

## 1. Introduction

Neglected Tropical Diseases (NTDs) are a group of heterogeneous illnesses causing severe health, economic, and social effects, primarily in the southern hemisphere [1]. Among them, the World Health Organization (WHO) has included Human African Trypanosomiasis (HAT), also known as sleeping sickness [2]. HAT is a vector-borne disease caused by two protozoa belonging to *Trypanosoma brucei* species, namely *T. b. gambiense* and *T. b. rhodesiense* [3]. Although the two subspecies are morphologically identical, they are differently widespread in Africa and determine a different clinical progression. *T. b. gambiense* is responsible for the chronic form of the disease (*g*HAT) and is mainly widespread in the West and Central African countries [4]. Meanwhile, *T. b. rhodesiense* causes the acute form of HAT (*r*HAT) in the region within the Horn of Africa and Zimbabwe [5]. The clinical course of HAT can be divided into two stages, depending on protozoa localization in the human body. After a tsetse fly bite, the protozoa enter the bloodstream, reaching mainly the spleen, lymph nodes, and spinal fluid, and causing non-specific symptoms, such as general malaise, headache, intermittent fever, and lymphadenopathy [4]. This first phase is known as stage 1 or hemolymphatic stage, which can last a few weeks or several years in the case of *r*HAT [6] and *g*HAT [7], respectively. Subsequently, the protozoa invade the central nervous system (CNS), leading to stage 2 (also known as the neurological stage), during which various mental disorders can occur, such as tremors, reversal of the normal sleep/wake cycle, motor disturbances, coma, and lastly death [8,9].

Despite the enormous efforts made by global health protection agencies and no-profit foundations, HAT still remains a health issue in rural regions and war zones [10]. Furthermore, many areas are difficult to reach and the recorded cases could be underestimated. Meanwhile, the possibility of moving easily from and to Africa exposes an uncontrolled number of people to the infection [11,12,13]. In the last century, only four drugs were available for the treatment of HAT. Suramin, pentamidine, melarsoprol, and eflornithine were largely used against *T. brucei* infections, but limited results were obtained [14,15]. Furthermore, a narrow spectrum of action, parenteral administration, and relevant toxicity, especially of melarsoprol, are associated with these outdated and unsafe drugs. At the beginning of the new millennium, the introduction of Nifurtimox-Eflornithine Combination Therapy (NECT) to the WHO Model List of Essential Medicines (EML) slightly improved the treatment of second stage *g*HAT [16]. Meanwhile, the very recent approval of fexinidazole, a new orally available nitroimidazole, has made a change in the treatment of HAT [17]. Fexinidazole has been shown to be effective against both stages of non-severe *g*HAT, but the arsenical derivative melarsoprol remains the drug of choice for the treatment of the neurological stage of *r*HAT [18]. In this context, considering the few available drugs, which moreover have a narrow spectrum of action, it appears clear that the development of new effective antitrypanosomal agents is still necessary.

Among all the possible strategies to arrest the most lethal *r*HAT, rhodesain, the main cysteine protease of *T. b. rhodesiense*, has emerged as a valid and precious target because of its key roles in the *Trypanosoma* life cycle [19]. This protozoal enzyme, also known as *Tbr*CatL [20], acts by means of the Cys25/His162/Asn182 catalytic triad, which is located in a pocket between the two domains of which the protease is composed. Rhodesain plays essential roles both in disease progression and the survival of the parasite: in fact, it is involved in the crossing of the blood–brain barrier, which promotes the neurological stage [21,22], and in the elusion of host immune response, since it is implicated in the formation of the variant surface glycoproteins (VSGs) expressed on the *Trypanosoma* coat [23,24]. In recent decades, several classes of potent peptide-based rhodesain inhibitors have been developed, while peptidomimetics and small molecules have also shown remarkable activity [25,26,27,28,29,30,31,32,33,34,35]. Unfortunately, although many ligands exhibited noteworthy inhibitory properties, most of them showed poor selectivity towards the target enzyme. Indeed, the inhibitory properties against human cathepsin L (hCatL), which is normally used for selectivity tests, have severely limited the drug-discovery process. In light of this, it appears clear that Structure–Activity Relationship (SAR) studies should aim to develop selective rhodesain inhibitors.

Among all the possible approaches, the introduction of unnatural amino acids and peptide backbone modifications has led to ligands with improved selectivity toward the target [36,37,38,39,40]. While the incorporation of an unnatural amino acid could be limited to a substitution, an extensive variety of backbone modifications could be inserted [41,42]. The introduction of peptoids, β- and γ-amino acids, azapeptides, *N*-methylation, and reduced peptide bonds are only some of the common modifications that may be incorporated along the peptide backbone [43,44,45,46,47,48].

In the last decades, our research team was actively involved in the synthesis of peptide-based and peptidomimetic rhodesain inhibitors [25,26,28,29,30,49,50,51,52]. In the docking studies performed on our most potent inhibitor, **PS-1** [50], no interactions between the ligand CO atom at the P3 site and the corresponding enzyme pocket were detected ([Fig molecules-27-03765-ch001]). Differently, the NH at the P2 site is predicted to be engaged in an H-bond interaction with Gly66. Meanwhile, the CO and NH at the P1 are involved in two H-bonds with the backbone of Gly66 and Asp161, respectively. Considering the role played by the backbone atoms in the interactions with the enzyme, only CO at the P3 site may be modified. Therefore, we decided to remove the carbonyl group, obtaining pseudopeptides with a reduced P2–P3 amide bond. The removal of the CO group could strongly change the chemical properties of the new molecules: indeed, pK_a_ calculations predict the secondary amine between the P3 and P2 sites to be easily protonated at physiological pH [53]. However, the presence of the secondary amine affects the vinyl ketone warhead of our most potent inhibitors: in fact, the R_2_-NH group could easily react with aldehydes and ketones providing carbinolamines, which spontaneously dehydrate to give enamines. In light of this, we decided to replace the methyl vinyl ketone warhead with the methyl ester analog, which is stable in the presence of secondary amines and proven to be a valid electron-withdrawing group (EWG) to activate the vinyl portion [49,54]. The Phe and hPhe residues at the P2 and P1 sites, respectively, were kept unchanged due to their fitting well into the corresponding enzyme pockets. The replacement of the phenyl ring at the P3 site with aliphatic and heteroaromatic substituents resulted in a relevant loss of affinity [50], the aromatic ring of **PS-1** was maintained, and various substituents were introduced. Besides the unsubstituted-phenyl ring (**SPR10**), both EWG and the electron-donating group (EDG) were introduced at the *para* position, providing the -NO_2_ (**SPR11**) and -OMe (**SPR12**) derivatives ([Fig molecules-27-03765-ch001]). According to our previously obtained best results [50,51], fluorine atoms were introduced, either one single atom (**SPR13**, **SPR14**, and **SPR15**) or two atoms (**SPR16**, **SPR17**, and **SPR18**), along with the -CF_3_ group at the *para* position (**SPR19**). 

In order to validate the role played by the P2–P3 reduced peptide bond, the methyl vinyl ester **PS-1** analogue (i.e., **SPR34**) was also developed.

In light of all these data, herein we report syntheses, biological evaluation, and docking studies of reduced peptide bond pseudopeptide Michael acceptors as potential anti-HAT agents.

## 2. Results

### 2.1. Synthesis

Synthesis of the desired compounds **SPR10**–**SPR19** was carried-out in solution following the “2+2” approach. In more detail, the hPhe-warhead (**8**) and *N*-benzyl-Phe (**4a–j**) fragments were separately synthesized and then merged in the last step by coupling reactions (Figure 1). The *N*-benzyl-Phe portions were synthesized by reductive amination starting from H-Phe-OMe **2** and the appropriate benzaldehydes **1a–j**, using sodium triacetoxyborohydride (STAB) as the reducing agent [55]. After the purification, the ester portion of intermediates **3a–j** was hydrolyzed with LiOH, and the inner salts **4a–j** were obtained after crystallization and re-crystallization with 1 M HCl and from methanol, respectively. On the other hand, the hPhe-warhead fragment **8** was synthesized starting from commercially available Boc-hPhe-OH, which was coupled with *N*,*O*-dimethylhydroxylamine chloride as recently reported [56], and the Weinreb amide **5** was reduced yielding the corresponding aldehyde **6** using LiAlH_4_. The methyl vinyl ester portion was inserted by Wittig reaction using methyl (triphenylphosphoranylidene)acetate as reagent, and the subsequent Boc-removal with 30% TFA in DCM provided the hPhe-warhead portion **8** as trifluoroacetate. Lastly, the coupling reactions between the hPhe-warhead portion and the different *N*-benzyl-Phe fragments **4a–j**, performed in the presence of TBTU and DIPEA, allowed us to obtain the final products, **SPR10**–**SPR19**, in moderate yields. Similarly to the pseudopeptides, **SPR34** was synthesized following the “2+2” approach: the hPhe-warhead fragment **8** was coupled with the P2–P3 pattern **9**, which was obtained as previously reported [50].

### 2.2. Biology

All molecules were tested towards rhodesain utilizing fluorogenic assays in the presence of Cbz-Phe-Arg-AMC as substrate. Considering that the methyl vinyl ester warhead irreversibly inhibits the enzyme target [49], the three classic inhibitory parameters, namely *k*_inact_, *K*_i_, and *k*_2nd_ were determined. Firstly, all the molecules were preliminarily screened at 100 µM (Figure 1A), and DMSO and E-64 were employed as the negative and positive control, respectively. Since **SPR10**–**SPR19** passed the initial screening (% of inhibition > 80%), they were properly diluted, in accordance with what is reported in the literature [57]. All the pseudopeptides showed low-micromolar and sub-micromolar *K*_i_ values, which represent the dissociation constant of the noncovalent enzyme-inhibitor complex [E·I] and reflect the affinity towards the target (Table 1). In more detail, the di-fluoro substituted analogs **SPR16**, **SPR17**, and **SPR18**, as well as the 4-CF_3_-containing derivative **SPR19**, exhibited comparable affinity in the submicromolar range against rhodesain. On the contrary, **SPR14**, **SPR15**, and **SPR16**, which carry only one fluorine atom at the *ortho*, *meta*, and *para* position, respectively, showed higher *K*_i_ values. The permanent presence of the fluorine atom at the *ortho* position in **SPR16**, **SPR17**, and **SPR18** moderately improved the affinity with respect to the mono-F-substituted inhibitors. For instance, the 2,4-F_2_ derivative **SPR16** showed a *K*_i_ value one order of magnitude better if compared with the 4-F analog **SPR15** (0.22 µM vs. 2.64 µM, respectively).

**SPR11**, which bears the nitro group at the *para* position, showed comparable activity to the disubstituted analogs, while the unsubstituted and 4-OMe analogs **SPR10** and **SPR12**, respectively, inhibited rhodesain poorly. Concerning the *k*_2nd_ values, the disubstituted **SPR16**, **SPR17**, and **SPR18**, the CF_3_-analogue **SPR19**, and the NO_2_-derivative **SPR11** showed the best results, with values between 1314 and 6950 M**^−^**^1^ min**^−^**^1^.

As can be easily noted, the amide bond reduction and the resulting presence of a secondary amine led to inhibitors with a lower affinity towards rhodesain with respect to the lead compound. On the other hand, **SPR34** showed potent rhodesain inhibition, with a *K*_i_ value in the low nanomolar range and *k*_2nd_ equal to 870,606 × 10^3^ M**^−^**^1^ min**^−^**^1^. The peptide inhibitor **SPR34**, which differs from **PS-1** solely in the warhead, exhibited a loss of affinity towards rhodesain with respect to that shown by the lead compound **PS-1**, indicating that the methyl vinyl ketone warhead better reacts with the catalytic cysteine with respect to the methyl vinyl ester portion. At the same time, the presence of the P2–P3 amide bond in **SPR34** provided better inhibitory properties if compared with those exhibited by the pseudopeptides, suggesting that the peptide backbone rigidity is better-tolerated by rhodesain with respect to the protonable reduced peptide bond -CH_2_-NH-.

In order to evaluate the selectivity towards the protozoan cysteine protease, all the new Michael acceptors were assayed against human Cathepsin L (hCatL), which represents the homologous protease of rhodesain in humans. With exception of **SPR11** and **SPR12**, the novel pseudopeptides showed only a low percentage of inhibition at the screening concentration (100 µM, Figure 1B), resulting in a significant selectivity towards the target (Table 1). The most potent rhodesain inhibitors, such as the di-fluoro-substituted analogs **SPR16**, **SPR17**, and **SPR18**, and the CF_3_-derivative **SPR19,** exhibited a very low hCatL inhibition, indicating that the reduced amide bond between the P3 and P2 sites is poorly tolerated by hCatL. Hence, if a significant loss of affinity towards rhodesain with respect to **PS-1** was observed (Table 1), similarly, the new Michael acceptors showed a meaningful selectivity when compared to the lead compound, which exhibited single-digit nanomolar activity against the human protease [50]. Similarly, **SPR34** showed a single-digit micromolar *K*_i_ value against hCatL, resulting in a Selectivity Index (SI) of 603 (*K*_i_ hCatL/rhodesain).

Lastly, the antitrypanosomal activity of the most potent analogs was evaluated against *T. b. brucei*, and the selected compounds showed single-digit micromolar EC_50_ values against the protozoa (Table 2, Figure 2A), comparable to those exhibited by the lead compound **PS-1** [50] and fexinidazole [58]. **SPR16** and **SPR18** exhibited EC_50_ values against the protozoa only one order of magnitude higher with respect to the rhodesain inhibition *K*_i_ values, while, as regards **PS-1**, a difference of six orders of magnitude between the values of EC_50_ against *T. b. brucei* (5.1 μM) and the *K*_i_ towards the enzyme target (1.1 pM) was observed. The great discrepancy in terms of rhodesain inhibition and the comparable antitrypanosomal activity between **PS-1** and the novel Michael acceptors might suggest an additional mechanism of action for the latter, for instance, the inhibition of other cysteine proteases, e.g., TbCatB [59]. The possibility of inhibiting two different targets represents one of the most promising strategies in the drug discovery process, as demonstrated by the high number of dual-inhibitors as antiprotozoal agents reported to date [60]. However, it should be considered that the removal of the carbonyl function in P3 turns the pseudopeptides into weak bases (predicted pK_a_ of the protonated species = 7.6–7.7 [53]), and therefore enables them to concentrate in acidic compartments [61,62], such as the protozoan lysosomes, where rhodesain is found. Despite the promising rhodesain inhibition, **SPR34** showed an EC_50_ value of 16.0 ± 2.1 µM against the protozoa, approximately 2 times higher with respect to those observed for **SPR16** and **SPR18**. This finding highlights that the presence of the reduced bond is crucial in significantly inhibiting protozoa growth.

In order to evaluate cytotoxicity, the three selected compounds were assayed against HEK293 cells. As can be noted in Figure 2B, **SPR16** and **SPR18** pseudopeptides, as well as the peptide inhibitor **SPR34**, showed no significant cytotoxicity at the tested concentrations towards mammalian cells. With respect to melarsoprol, which is considered the drug of choice of severe *r*HAT, the pseudopeptides reported herein were shown to be better-tolerated in mammalian cells: in fact, melarsoprol showed an IC_50_ value of 3.3 µM towards L-6 rat myoblast cells [63], while a lack of relevant cytotoxicity up to 70 µM towards HEK293 cells was observed for **SPR16**, **SPR18**, and **SPR34**.

Considering the high degree of homology between cysteine proteases in Trypanosomatids, we can speculate regarding the potential use of the rhodesain inhibitors reported herein against a panel of *Trypanosoma* species infecting animals [64]. To name a few, *T. congolense*, *T. vivax*, *T. evansi*, *T. avium*, and *T. equiperdum* could infect several species of animals causing alteration of the ecosystems, killing the animals involved, and requiring a serious commitment of human and financial resources [65]. For instance, *T. congolese* possesses congopain, a lysosomal enzyme which shares a high degree of homology with rhodesain [66]. In light of all of these, the antitrypanosomal activity of SPRs described above towards a panel of protozoa belonging to the *Trypanosomatidae* family could be investigated.

### 2.3. Molecular Modeling

To gain insight into the molecular interactions between the newly discovered compounds and rhodesain, molecular-docking experiments were performed on compounds **SPR16** and **SPR18**, which are the most potent antitrypanosomal agents of the series. These two ligands were covalently docked to the C25 in the rhodesain active site [67] following the covalent docking protocol available in Glide [68]. Figure 3 reports the predicted docking solution achieved for the two compounds (Figure 3A,B). In particular, in both cases, the carbonyl oxygen of the ester warhead forms a double H-bond with the Q19 and W184 side chains, the P1 residue engages an additional H-bond with D161 (backbone CO) through its backbone NH, and the P2 carbonyl oxygen and protonated amine form a double bridged H-bond with G66 backbone NH and CO, respectively. In particular, while the carbonyl oxygen forms a H-bond with the backbone NH of G66 (distance between the hydrogen atom and the oxygen 1.80 Å), the ligand protonated amine is able to form a charge-reinforced H-bond with the backbone CO of the same residue (distance between the hydrogen atom and the oxygen 2.12 Å). As for the ligands’ side chains, the P2 phenyl ring is positioned in the lipophilic S2 cleft, which comprises the residues M68, A138, L160, and A208.

On the other hand, the P3 side chains of both ligands are placed in the S3 cleft forming positive van der Waals interactions alternatively with Q159 (**SPR16**) or F61 (**SPR18**). These contacts are reinforced by the electron-withdrawing effect of the fluorine atoms decorating the P3 phenyl ring. All in all, the novel compounds bind rhodesain consistently, with the binding poses already achieved for other structural congeners [49,50,51,52]. Nevertheless, the achieved theoretical results are unable to clearly indicate the reasons for the lower inhibitory activity with respect to the parent compound **PS-1** [50,67]. In this respect, we speculate that the lower inhibitory efficiency might be ascribed to the less pronounced electrostatic complementarity between the positively charged ligands and the lipophilic enzyme active site.

To clarify the novel inhibitors′ selectivity profile between rhodesain and the human cathepsin L, compounds **SPR16** and **SPR18** were also covalently docked to the reactive C26 in the cathepsin L active site, following the same docking protocol employed for the parasitic protease. Unfortunately, Glide failed in providing a possible solution as no low-energy poses were found. To rationalize this result, the complexes achieved when ligands were docked into the rhodesain binding site were rigidly superimposed on the human cathepsin L X-ray structure (PDB ID: 3OF9-human cathepsin L in complex with a diazomethylketone inhibitor [69]; Figure 4A,B). From this analysis, it is possible to speculate that the differences in the residues lining the S2 and S3 clefts might be the reasons for the selectivity of the inhibitors. In particular, in the S3 cleft, the rhodesain residues F61 and Q159 are replaced in cathepsin L by E64 and D161, respectively. As mentioned earlier, in rhodesain, the P3 phenyl ring can form favorable interactions with F61 or Q159, while in cathepsin L, the same aromatic ring might be repulsed by the negatively charged E64 or D161 residues. This repulsion is exacerbated by the partially negative fluorine substituents. Additionally, it could be hypothesized that the specific substitution pattern of **SPR16** and **SPR18** makes them also less prone to establishing favorable contacts with cathepsin L while approaching the enzyme active site. This could explain why the presence of strong H-bond acceptors in the *para* position of the P3 phenyl ring rescues the ability to inhibit cathepsin L (**SPR11** and **SPR12** in Table 1).

## 3. Materials and Methods

### 3.1. Chemistry

All reagents and solvents were purchased from commercial suppliers. *N*,*O*-dimethylhydroxylamine, TBTU, and DIPEA were obtained from Fluorochem. The benzaldehydes, TEA, AcOH, TFA, STAB, H-Phe-OH, LiOH, LiAlH_4_, and methyl (triphenylphosphoranylidene)acetate were obtained from Merck, as well as silica gel 60 F254 plates and silica gel (200–400 mesh) employed for TLC and column chromatography, respectively. All the ^1^H and ^13^C spectra were performed on a Varian 500 MHz provided with a ONE NME probe operating at 499.74 and 125.73 MHz for ^1^H and ^13^C, respectively. Deuterated solvents (i.e., CDCl_3_, MeOD, and D_2_O) were obtained from Merck and the signal of the solvents was used as the internal standard. Splitting patterns are described as singlet (s), doublet (d), doublet of doublet (dd), triplet (t), quartet (q), multiplet (m), and broad singlet (bs). Chemical shifts are expressed in ppm and coupling constants (J) in Hz. Elemental analyses were performed on a C. Erba model 1106 (elemental analyzer for C, H, and N) apparatus, and ±0.4% of the theoretical values were found.

#### 3.1.1. General Procedure Followed for the Synthesis of Intermediates **3a–j**

In a round bottom flask, the appropriate benzaldehyde (1 eq) was dissolved in DCE (15 mL/mmol), glacial AcOH (1 eq) was added and the reaction was maintained by vigorously stirring for 2 h at rt. After this time, H-Phe-OH (1 eq.) was added and the reaction was maintained in stirring for a further 2 h. After that, STAB (1.5 eq) was added portion-wise over a period of 20 min, and the reaction was kept in stirring for 48 h. STAB (1 eq) was further added portion-wise and the reaction was maintained in stirring for a further 2 h. After this time, 5% of NaHCO_3_ solution (20 mL/mmol) was added and DCE was evaporated. The organic phase was extracted with EtOAc (20 mL × 3), dried over Na_2_SO_4_ and concentrated in vacuo. All the esters were purified with the appropriate eluent mixture described below in detail.

*Methyl benzyl-L-phenylalaninate* (**3a**) Column chromatography in light petroleum/EtOAc (9:1). *R*_f_ = 0.41 (light petroleum/EtOAc, 9:1). Consistency: white powder. Yield = 74%. ^1^H NMR (500 MHz) in CDCl_3_, *δ* = 2.94 (dd, *J* = 11.5, 5.1 Hz, 1H), 2.98 (dd, *J* = 11.5, 4.5 Hz, 1H), 3.54 (t, *J* = 6.9 Hz, 1H), 3.63 (d, *J* = 13.2 Hz, 1H), 3.64 (s, 3H), (d, *J* = 13.2 Hz, 1H), 7.13–7.18 (m, 2H), 7.19–7.3 (m, 8H). ^13^C NMR (126 MHz) in CDCl_3_, *δ* = 39.70, 51.59, 51.97, 62.02, 126.67, 127.02, 128.13, 128.32, 128.36, 129.21, 137.28, 139.51, 174.95. NMR data are in agreement with those reported in the literature [70].

*Methyl (4-nitrobenzyl)-L-phenylalaninate* (**3b**) Column chromatography in light petroleum/EtOAc (4:1). *R*_f_ = 0.29 (light petroleum/EtOAc, 4:1). Consistency: pale yellow powder. Yield = 72%. ^1^H NMR (500 MHz) in CDCl_3_, *δ* = 2.91 (dd, *J* = 13.5, 7.7 Hz, 1H), 3.01 (dd, *J* = 13.6, 5.9 Hz, 1H), 3.46 (dd, *J* = 7.7, 5.9 Hz, 1H), 3.68 (s, 3H), 3.69 (d, *J* = 14.6 Hz, 1H), 3.93 (d, *J* = 14.6 Hz, 1H), 7.11–7.18 (m, 2H), 7.21–7.30 (m, 3H), 7.33 (d, *J* = 8.8 Hz, 2H), δ 8.07 (d, *J* = 8.8 Hz, 2H). ^13^C NMR (125 MHz) in CDCl_3_, *δ* = 39.77, 51.17, 51.91, 62.11, 123.53, 126.88, 128.48, 128.61, 129.31, 137.18, 147.08, 147.45, 174.85. NMR data are in agreement with those reported in the literature [71].

*Methyl (4-methoxybenzyl)-L-phenylalaninate* (**3c**) Column chromatography in light petroleum/EtOAc (4:1). *R*_f_ = 0.40 (light petroleum/EtOAc, 4:1). Consistency: white powder. Yield = 84%. ^1^H NMR (500 MHz) in CDCl_3_, *δ* = 2.89–2.98 (m, 2H), 3.52 (t, *J* = 6.9 Hz, 1H), 3.55 (d, *J* = 12.9 Hz, 1H), 3.60 (s, 3H), 3.72 (d, *J* = 13.1 Hz, 1H), 3.72 (s, 3H), 6.77–6.81 (m, 2H), 7.10–7.15 (m, 4H), 7.16–7.21 (m, 1H), 7.21–7.25 (m, 2H). ^13^C NMR (126 MHz) in CDCl_3_, *δ* = 39.61, 51.30, 51.48, 55.09, 61.83, 113.64, 126.57, 128.27, 129.13, 129.26, 131.55, 137.26, 158.62, 174.91. NMR data are in agreement with those reported in the literature [72].

*Methyl (2-fluorobenzyl)-L-phenylalaninate* (**3d**) Column chromatography in light petroleum/EtOAc (9:1). *R*_f_ = 0.33 (light petroleum/EtOAc, 9:1). Consistency: white powder. Yield = 70%. ^1^H NMR (500 MHz) in CDCl_3_, *δ* = 2.94 (dd, *J* = 12.6, 6.3 Hz, 1H), 2.97 (dd, *J* = 12.6, 5.6 Hz, 1H), 3.53 (t, *J* = 6.9 Hz, 1H), 3.59 (s, 3H), 3.72 (d, *J* = 13.9 Hz, 1H), 3.83 (d, *J* = 13.9 Hz, 1H), 6.94–6.98 (m, 1H), 7.02 (td, *J* = 7.5, 1.1 Hz, 1H), 7.09–7.30 (m, 7H). ^13^C NMR (126 MHz) in CDCl_3_, *δ* = 39.66, 45.32 (d, *J* = 3.4 Hz), 51.65, 62.15, 115.15 (d, *J* = 21.7 Hz), 123.99 (d, *J* = 3.6 Hz), 126.50 (d, *J* = 14.7 Hz), 126.71, 128.40, 128.69 (d, *J* = 8.1 Hz), 129.18, 130.18 (d, *J* = 4.5 Hz), 137.20, 161.09 (d, *J* = 245.8 Hz), 174.75.

*Methyl (3-fluorobenzyl)-L-phenylalaninate* (**3e**) Column chromatography in light petroleum/EtOAc (9:1). *R*_f_ = 0.32 (light petroleum/EtOAc, 9:1). Consistency: white powder. Yield = 67%. ^1^H NMR (500 MHz) in CDCl_3_, *δ* = 2.93 (dd, *J* = 13.5, 7.3 Hz, 1H), 2.97 (dd, *J* = 13.5, 6.4 Hz, 1H), 3.50 (dd, *J* = 7.3, 6.4 Hz, 1H), 3.60 (d, *J* = 13.7 Hz, 1H), 3.63 (s, 3H), 3.80 (d, *J* = 13.7 Hz, 1H), 6.85–7.00 (m, 3H), 7.10–7.23 (m, 4H), 7.24–7.30 (m, 2H). ^13^C NMR (126 MHz) in CDCl_3_, *δ* = 39.73, 51.38 (d, *J* = 1.8 Hz), 51.68, 61.96, 113.86 (d, *J* = 21.2 Hz), 114.84 (d, *J* = 21.5 Hz), 123.56 (d, *J* = 2.8 Hz), 126.77, 128.41, 129.24, 129.72 (d, *J* = 8.2 Hz), 137.22, 142.36 (d, *J* = 7.0 Hz), 162.94 (d, *J* = 245.6 Hz), 174.91.

*Methyl (4-fluorobenzyl)-L-phenylalaninate* (**3f**) Column chromatography in light petroleum/EtOAc (9:1). *R*_f_ = 0.35 (light petroleum/EtOAc, 9:1). Consistency: white powder. Yield = 57%. ^1^H NMR (500 MHz) in CDCl_3_, *δ* = 2.94 (qd, *J* = 13.5, 6.9 Hz, 1H), 3.49 (dd, *J* = 7.3, 6.5 Hz, 1H), 3.57 (d, *J* = 13.3 Hz, 1H), 3.63 (s, 3H), 3.76 (d, *J* = 13.3 Hz, 1H), 6.90–9.95 (m, 2H), 7.15 (dt, *J* = 8.1, 4.6 Hz, 4H), 7.18–7.23, (m,1H), 7.26 (tt, *J* = 8.0, 1.7 Hz, 2H). ^13^C NMR (126 MHz) in CDCl_3_, *δ* = 39.73, 51.21, 51.65, 61.94, 115.07 (d, *J* = 21.3 Hz), 126.72, 128.39, 129.23, 129.67 (d, *J* = 7.9 Hz), 135.30 (d, *J* = 3.1 Hz), 137.30, 161.95 (d, *J* = 244.6 Hz), 174.96.

*Methyl (2,4-difluorobenzyl)-L-phenylalaninate* (**3g**) Column chromatography in light petroleum/EtOAc (9:1). *R*_f_ = 0.24 (light petroleum/EtOAc, 9:1). Consistency: white powder. Yield = 44%. ^1^H NMR (500 MHz) in CDCl_3_, *δ* = 2.92 (dd, *J* = 13.6, 7.4 Hz, 1H), 2.98 (dd, *J* = 13.6, 6.3 Hz, 1H), 3.50 (dd, *J* = 7.4, 6.4 Hz, 1H), 3.63 (s, 3H), 3.67 (d, *J* = 13.9 Hz, 1H), 3.78 (d, *J* = 13.9 Hz, 1H), 6.68–6-79 (m, 2H), 7.12–7.16 (m, 2H), 7.17–7.12 (m, 1H), 7.24–7.29 (m, 2H). ^13^C NMR (126 MHz) in CDCl_3_, *δ* = 39.65, 44.75 (d, *J* = 2.9 Hz), 51.68, 62.05, 103.55 (t, *J* = 25.5 Hz), 110.94 (dd, *J* = 20.9, 3.7 Hz), 122.48 (dd, *J* = 14.9, 3.7 Hz), 126.72, 128.39, 129.15, 130.87 (dd, *J* = 9.6, 6.3 Hz), 137.17, 160.52 (dd, *J* = 138.2, 11.9 Hz), 162.49 (dd, *J* = 137.2, 11.9 Hz), 174.72.

*Methyl (2,5-difluorobenzyl)-L-phenylalaninate* (**3h**) Column chromatography in light petroleum/EtOAc (9:1). *R*_f_ = 0.26 (light petroleum/EtOAc, 9:1). Consistency: white powder. Yield = 56%. ^1^H NMR (500 MHz) in CDCl_3_, *δ* = 2.93 (dd, *J* = 13.6, 7.5 Hz, 1H), 3.00 (dd, *J* = 13.6, 6.2 Hz, 1H), 3.50 (dd, *J* = 7.4, 6.2 Hz, 1H), 3.64 (s, 3H), 3.67 (d, *J* = 14.5 Hz, 1H), 3.81 (d, *J* = 14.6 Hz, 1H), 6.81–6.87 (m, 1H), 6.88–6.91 (m, 1H), 6.92–6.97 (m, 1H), 7.13–7.19 (m, 1H), 7.19–7.25 (m, 1H), 7.25–7.31 (m, 1H). ^13^C NMR (126 MHz) in CDCl_3_, *δ* = 39.67, 44.81 (dd, *J* = 3.0, 0.9 Hz), 51.72, 62.11, 114.68 (dd, *J* = 24.2, 8.6 Hz), 116.08 (ddd, *J* = 24.8, 21.4, 6.8 Hz), 126.79, 128.42, 129.17, 137.09, 156.77 (dd, *J* = 237.4, 2.3 Hz), 158.69 (dd, *J* = 238.0, 2.4 Hz), 174.69.

*Methyl (2,6-difluorobenzyl)-L-phenylalaninate* (**3i**) Column chromatography in light petroleum/EtOAc (9:1). *R*_f_ = 0.28 (light petroleum/EtOAc, 9:1). Consistency: white powder. Yield = 75%. ^1^H NMR (500 MHz) in CDCl_3_, *δ* = 2.92 (dd, *J* = 12.5, 6.1 Hz, 1H), 2.96 (dd, *J* = 12.5, 5.5 Hz, 1H), 3.53 (t, *J* = 7.0 Hz, 1H), 3.55 (s, 3H), 3.82 (d, *J* = 13.4 Hz, 1H), 3.87 (d, *J* = 13.4 Hz, 1H), 6.77–6.84 (m, 2H), 7.09–7.14 (m, 2H), 7.14–7.21 (m, 2H), 7.21–7.26 (m, 2H). ^13^C NMR (126 MHz) in CDCl_3_, *δ* = 39.21 (t, *J* = 3.0 Hz), 39.51, 51.64, 62.01, 111.14 (dd, *J* = 20.2, 6.0 Hz), 115.12 (t, *J* = 20.0 Hz), 126.70, 128.40, 129.04, 129.04 (s), 137.01, 161.70 (dd, *J* = 248.0, 8.5 Hz), 174.46.

*Methyl (4-(trifluoromethyl)benzyl)-L-phenylalaninate* (**3j**) Column chromatography in light petroleum/EtOAc (9:1). *R*_f_ = 0.32 (light petroleum/EtOAc, 9:1). Consistency: white powder. Yield = 58%. ^1^H NMR (500 MHz) in CDCl_3_, *δ* = 2.92 (dd, *J* = 13.5, 7.5 Hz, 1H), 2.99 (dd, *J* = 13.6, 6.2 Hz, 1H), 3.49 (dd, *J* = 7.5, 6.2 Hz, 1H), 3.65 (s, 3H), 3.65 (d, *J* = 14.1 Hz, 1H), 3.87 (d, *J* = 14.0 Hz, 1H), 7.14–7.18 (m, 2H), 7.20–7.25 (m, 1H), 7.25–7.31 (m, 4H), 7.49 (d, *J* = 8.1 Hz, 2H). ^13^C NMR (126 MHz) in CDCl_3_, *δ* = 39.81, 51.46, 51.77, 62.07, 124.34 (q, *J* = 271.9 Hz), 125.27 (q, *J* = 3.8 Hz), 126.84, 128.32, 128.48, 129.30 (q, *J* = 32.2 Hz), 129.32, 137.28, 143.84, 174.95.

#### 3.1.2. General Procedures for the Synthesis of Inner Salts **4a–j**

In a round bottom flask, the appropriate ester **3a–j** (1 eq) was dissolved in MeOH and dioxane (3.3 mL for each/mmol). The flask was inserted in an ice-bath and LiOH as a fine powder (2 eq) and H_2_O (3.3 mL/mmol) were added, and the reaction was maintained in stirring until the disappearance of the starting material (monitoring in TLC using the same eluent mixture for the ester purification). After that, volatiles were removed in vacuo, water was added and the pH was adjusted to 1 using 1 M HCl solution, and the desired compounds were crystallized at low temperature along with LiCl. The subsequent recrystallization from methanol provided the title inner salts, which were used for the next step without further purification.

*(S)-2-(benzylammonio)-3-phenylpropanoate* (**4a**) ^1^H NMR (500 MHz) in D_2_O, *δ* = 2.83 (d, *J* = 6.8 Hz, 2H), 3.28 (t, *J* = 6.9 Hz, 1H), 3.46 (d, *J* = 12.8 Hz, 1H), 3.64 (d, *J* = 12.8 Hz, 1H), 7.10–7.31 (m, 10H). ^13^C NMR (126 MHz) in D_2_O, *δ* = 38.93, 50.93, 64.24, 126.73, 127.48, 128.24, 128.71, 128.96, 129.39, 138.00, 138.74, 180.98 [73].

*(S)-2-((4-nitrobenzyl)ammonio)-3-phenylpropanoate* (**4b**) ^1^H NMR (500 MHz) in D_2_O, *δ* = 2.88 (dd, *J* = 11.8, 4.9 Hz, 1H), 2.92 (dd, *J* = 11.8, 5.1 Hz, 1H), 3.32 (t, *J* = 6.9 Hz, 1H), 3.65 (d, *J* = 13.8 Hz, 1H), 3.83 (d, *J* = 13.7 Hz, 1H), 7.22 (d, *J* = 8.0 Hz, 2H), 7.24–7.29 (m, 1H), 7.30–7.35 (m, 2H), 7.39 (d, *J* = 7.8 Hz, 2H), 8.10 (d, *J* = 7.6 Hz, 2H). ^13^C NMR (126 MHz) in D_2_O, *δ* = 38.57, 49.92, 64.10, 122.99, 123.22, 126.04, 127.99, 128.70, 137.51, 146.06, 146.37, 180.36.

*(S)-2-((4-methoxybenzyl)ammonio)-3-phenylpropanoate* (**4c**) ^1^H NMR (500 MHz) in D_2_O, *δ* = 2.76 (dd, *J* = 13.5, 6.9 Hz, 1H), 2.82 (dd, *J* = 13.2, 6.8 Hz, 1H), 3.24 (t, *J* = 6.9 Hz, 1H), 3.34 (d, *J* = 12.4 Hz, 1H), 3.52 (d, *J* = 12.6 Hz, 1H), 3.57 (s, 3H), 6.72 (d, *J* = 8.6 Hz, 2H), 7.02 (d, *J* = 7.5 Hz, 2H), 7.13–7.07 (m, 3H), 7.17 (t, *J* = 7.2 Hz, 2H). ^13^C NMR (126 MHz) in D_2_O, *δ* = 38.89, 50.45, 54.95, 63.95, 113.61, 126.58, 128.61, 129.28, 129.88, 131.39, 138.07, 157.71, 180.95.

*(S)-2-((2-fluorobenzyl)ammonio)-3-phenylpropanoate* (**4d**) ^1^H NMR (500 MHz) in D_2_O, *δ* = 2.93–2.82 (m, 2H), 3.33 (t, *J* = 6.9 Hz, 1H), 3.60 (d, *J* = 13.1 Hz, 1H), 3.74 (d, *J* = 13.1 Hz, 1H), 7.10–7.04 (m, 1H), 7.12 (t, *J* = 7.4 Hz, 1H), 7.20 (d, *J* = 7.1 Hz, 1H), 7.24 (dd, *J* = 10.3, 4.1 Hz, 1H), 7.33–7.27 (m, 1H). ^13^C NMR (126 MHz) in D_2_O, *δ* = 38.97, 44.60, 64.52, 115.27 (d, *J* = 20.2 Hz), 124.38 (d, *J* = 3.5 Hz), 125.46 (d, *J* = 15.2 Hz), 126.50, 128.69, 129.35, 129.51 (d = *J* = 9.4 Hz), 131.04 (d, *J* = 3.3 Hz), 137.96, 160.84 (d, *J* = 244.1 Hz), 180.85.

*(S)-2-((3-fluorobenzyl)ammonio)-3-phenylpropanoate* (**4e**) ^1^H NMR (500 MHz) in D_2_O, *δ* = 2.86 (d, *J* = 6.8 Hz, 2H), 3.30 (td, *J* = 6.9, 1.2 Hz, 1H), 3.52 (d, *J* = 13.1 Hz, 1H), 3.70 (d, *J* = 12.4 Hz, 1H), 6.95–7.06 (m, 3H), 7.16–7.21 (m, 2H), 7.22–7.27 (m, 1H), 7.27–7.33 (m, 3H). ^13^C NMR (126 MHz) in D_2_O, *δ* = 36.77, 48.27, 62.18, 111.76 (d, *J* = 29.0 Hz), 112.87 (d, *J* = 24.6 Hz), 124.66, 125.91 (d, *J* = 3.8 Hz), 126.57, 127.29, 128.24 (d, *J* = 8.4 Hz), 135.76, 139.10 (d, *J* = 12.2 Hz), 160.25 (d, *J* = 243.1 Hz), 178.74.

*(S)-2-((4-fluorobenzyl)ammonio)-3-phenylpropanoate* (**4f**) ^1^H NMR (500 MHz) in D_2_O, *δ* = 2.70 (d, *J* = 6.7 Hz, 2H), 3.14 (dt, *J* = 6.7, 3.3 Hz, 1H), 3.31 (d, *J* = 12.8 Hz, 1H), 3.49 (d, *J* = 12.7 Hz, 1H), 6.85 (t, *J* = 8.3 Hz, 2H), 6.99–7.09 (m, 5H), 7.09–7.15 (m, 2H). ^13^C NMR (126 MHz) in D_2_O, *δ* = 38.93, 50.23, 64.15, 114.98 (d, *J* = 18.5 Hz), 128.13, 128.69, 129.37, 130.44 (d, *J* = 9.3 Hz), 134.57 (d, *J* = 2.4 Hz), 138.00, 161.64 (d, *J* = 242.4 Hz), 180.96.

*(S)-2-((2,4-difluorobenzyl)ammonio)-3-phenylpropanoate* (**4g**) ^1^H NMR (500 MHz) in D_2_O, *δ* = 2.95–2.83 (m, 2H), 3.35 (t, *J* = 6.9 Hz, 1H), 3.62 (d, *J* = 13.1 Hz, 1H), 3.75 (d, *J* = 13.1 Hz, 1H), 6.94 (dd, *J* = 9.8, 8.7 Hz, 2H), 7.21–7.25 (m, 2H), 7.25–7.32 (m, 2H), 7.32–7.37 (m, 2H). ^13^C NMR (126 MHz) in D_2_O, *δ* = 39.00, 44.14, 64.50, 103.48 (t, *J* = 26.1 Hz), 111.01 (d, *J* = 23.2 Hz), 121.58 (dd, *J* = 15.8, 3.4 Hz), 126.78, 128.72, 129.39, 133.48 (t, *J* = 7.4 Hz), 137.93, 161.41 (dd, *J* = 87.5, 5.3 Hz), 162.45 (dd, *J* = 143.6, 5.1 Hz), 180.85.

*(S)-2-((2,5-difluorobenzyl)ammonio)-3-phenylpropanoate* (**4h**) ^1^H NMR (500 MHz) in D_2_O, *δ* = 2.75 (d, *J* = 6.8 Hz, 2H), 3.20 (td, *J* = 6.9, 1.2 Hz, 1H), 3.46 (d, *J* = 13.5 Hz, 1H), 3.59 (d, *J* = 13.4 Hz, 1H), 6.96 (m, 3H), 7.06–7.10 (m, 2H), 7.10–7.15 (m, 1H), 7.16–7.22 (m, 2H). ^13^C NMR (126 MHz) in D_2_O, *δ* = 38.99, 44.39, 64.51, 115.28 (dd, *J* = 27.9, 21.3 Hz), 116.71 (dd, *J* = 27.4, 6.6 Hz), 126.79, 127.20 (dd, *J* = 18.4, 7.9 Hz), 128.73, 129.38, 137.93, 156.57 (d, *J* = 175.7 Hz), 158.48 (d, *J* = 175.6 Hz), 180.76.

*(S)-2-((2,6-difluorobenzyl)ammonio)-3-phenylpropanoate* (**4i**) ^1^H NMR (500 MHz) in D_2_O, *δ* = 2.70–2.81 (m, 2H), 3.24 (t, *J* = 6.6 Hz, 2H), 3.60 (d, *J* = 13.0 Hz, 1H), 3.68 (d, *J* = 13.0 Hz, 1H), 6.84 (t, *J* = 7.4 Hz, 2H), 7.05–7.12 (m, 2H), 7.12–7.25 (m, 3H). ^13^C NMR (126 MHz) in D_2_O, *δ* = 38.28, 39.05, 64.64, 111.16 (dd, *J* = 20.2, 5.7 Hz), 114.16 (t, *J* = 19.4 Hz), 126.51, 128.46, 129.12, 129.60 (t, *J* = 10.7 Hz), 137.95, 161.29 (dd, *J* = 246.1, 9.1 Hz), 180.71.

*(S)-3-phenyl-2-((4-(trifluoromethyl)benzyl)ammonio)propanoate* (**4j**) ^1^H NMR (500 MHz) in acetone-d6, *δ* = 2.90 (dd, *J* = 13.5, 10.2 Hz, 1H), 3.26 (dd, *J* = 13.5, 3.2 Hz, 1H), 3.37 (dd, *J* = 9.9, 3.6 Hz, 1H), 3.53 (d, *J* = 14.4 Hz, 1H), 3.90 (d, *J* = 14.4 Hz, 1H), 7.04 (d, *J* = 7.9 Hz, 2H), 7.10–1.17 (m, 3H), 7.19 (d, *J* = 6.3 Hz, 2H), 7.27 (d, *J* = 8.1 Hz, 2H). ^13^C NMR (126 MHz) in acetone-d6, δ = 40.27, 52.13, 65.26, 125.20 (q, *J* = 271.0 Hz), 125.52 (dd, *J* = 21.8, 6.2 Hz), 126.86, 128.97 (dd, *J* = 21.8, 18.3 Hz), 129.02, 129.14, 130.19 (q, *J* = 32.0 Hz), 140.14, 145.51, 182.00.

#### 3.1.3. Synthesis of hPhe-Warhead Fragment **8**

*(S)-tert-butyl (1-(methoxyamino)-1-oxo-4-phenylbutan-2-yl)carbamate* (**5**) The Weinreb amide **5** was obtained as we have recently reported [56].

*(S)-tert-butyl (1-oxo-4-phenylbutan-2-yl)carbamate* (**6**) The Weinreb amide **5** was dissolved in dry THF (2.5 mL/mmol) in N_2_ atmosphere, and the flask was cooled to −10/−15 °C using an ice/bath. LiAlH_4_ (1 eq) was added portion-wise over a period of 40 min, under vigorous stirring. The reaction was maintained in stirring for a further 2 h at no more than −5 °C. If TLC monitoring confirmed the disappearance of the starting material, 1M KHSO_4_ was carefully added to quench LiAlH_4_. After that, Et_2_O was added and the aqueous layer was separated and extracted with Et_2_O (×3). The combined organic layers were washed with 1M HCl (×2), water (×2), and brine (×2), dried over Na_2_SO_4_ and concentrated in vacuo. The desired intermediate was purified by column chromatography using light petroleum/EtOAc (4:1) as eluent mixture. Consistency = colorless solid. *R*_f_ = 0.52 (light petroleum/EtOAc, 4:1), Yield = 82%. ^1^H NMR (500 MHz) in CDCl_3_, *δ* = 1.46 (s, 9H) 1.84–1.92 (m, 1H), 2.17–2.29 (m, 1H), 2.67–2.74 (m, 2H), 4.19–4.30 (m, 1H),5.10 (bs, 1H), 7.16–7.24 (m, 3H), 7.26–7.31 (m, 2H), 9.55 (s, 1H). ^13^C NMR (500 MHz) in CDCl_3_, *δ* = 28.29, 30.82, 31.42, 59.52, 80.13, 126.32, 128.40, 128.58, 140.59, 155.55, 199.62. NMR data are in agreement with those reported in the literature [74].

*(S,E)-methyl 4-((tert-butoxycarbonyl)amino)-6-phenylhex-2-enoate* (**7**) In a round bottom flask, the aldehyde **6** (1 eq) was dissolved in DCM (3 mL/mmol) and the Wittig reagent methyl (triphenylphosphoranylidene)acetate (1.1 eq) was added at rt. The reaction was vigorously stirred for 2 h. After this time, the solvent was removed in vacuo and the desired product was obtained by column chromatography using light petroleum/EtOAc (9:1) as the eluent mixture. Consistency = white solid. *R*_f_ = 0.39 (light petroleum/EtOAc, 9:1), Yield = 86%. ^1^H NMR (500 MHz) in CDCl_3_, *δ* = 1.44 (s, 9H), 1.72–1.94 (m, 2H), 2.56–2.75 (m, 2H), 3.71 (s, 3H), 4.23–4.38 (m, 1H), 4.81–4.95 (m, 1H), 5.93 (dd, *J* = 15.7, 1.2 Hz, 1H), 6.87 (dd, *J* = 15.3, 5.3 Hz, 1H), 7.12–7.21 (m, 3H), 7.22–7.29 (m, 2H). ^13^C NMR (500 MHz) in CDCl_3_, *δ* = 28.34, 32.02, 36.16, 51.22, 51.57, 79.65, 120.43, 126.10, 128.33, 128.48, 140.92, 148.62, 155.17, 166.73.

*(S,E)-6-methoxy-6-oxo-1-phenylhex-4-en-3-aminium 2,2,2-trifluoroacetate* (**8**) In a round bottom flask, the intermediate **7** (1 eq) was dissolved in DCM at rt, TES (0.01 eq) and 30% of TFA (10 eq) were added dropwise. The reaction was kept in stirring until the disappearance of SM (TLC monitoring = light petroleum/EtOAc, 9:1, 2–3 h). After that, DCM and TFA were removed in vacuo using toluene (×3) and CHCl_3_ (×1) and the trifluoroacetate aminium salt was triturated in Et_2_O, and used for the next step without purification. Consistency = white powder; Yield = 98%; ^1^H NMR (500 MHz) in MeOD, *δ* = 1.97–2.08 (m, 1H), 2.08–2.17 (m, 1H), 2.57–2.76 (m, 2H), 3.78 (s, 3H), 3.90–3.99 (m, 1H), 6.17 (dd, *J* = 15.8, 0.8 Hz, 1H), 6.86 (dd, J = 15.8, 7.9 Hz, 1H), 7.16–7.25 (m, 3H), 7.25–7.33 (m, 2H). ^13^C NMR (500 MHz) in MeOD, *δ* = 32.27, 35.48, 52.49, 52.93, 126.63, 127.51, 129.36, 129.71, 141.25, 142.86, 167.01.

#### 3.1.4. General Procedure for the Synthesis of Final Products **SPR10**–**SPR19** and **SPR34**

In a round bottom flask (A), the appropriate inner salt, **4a–j** or **9** [50] (1.2 eq), was suspended in DCM (15 mL/mmol) at 0 °C, and TBTU (1.2 eq) and DIPEA (1.5 eq) were added. The reaction was maintained in stirring for 20 min. In a second flask (B), the trifluoroacetate salt **8** (1 eq) was suspended in DCM (15 mL/mmol) at 0 °C, DIPEA was added (2 eq), and the Ph was checked (>7). After 10 min, the amine (B) was added dropwise to the flask (A), Ph was checked (>7), and the reaction was kept in stirring at rt. Following this, DCM was removed *in vacuo* and the resulting residue was dissolved in EtOAc, washed with brine (×3), dried over Na_2_SO_4_, and concentrated in vacuo. The desired product was purified by column chromatography using the appropriate eluent mixture reported below.

*Methyl (S,E)-4-((S)-2-(benzylamino)-3-phenylpropanamido)-6-phenylhex-2-enoate* (**SPR10**) Column chromatography in light petroleum/EtOAc (7:3). Consistency = pale yellow powder. *R*_f_ = 0.42 (light petroleum/EtOAc, 7:3). Yield = 47%. ^1^H NMR (500 MHz) in CDCl_3_, *δ* = 1.78–1.89 (m, 1H), 1.89–1.97 (m, 1H), 2.61 (t, *J* = 7.9 Hz, 2H), 2.77 (dd, *J* = 13.8, 9.0 Hz, 1H), 3.20 (dd, *J* = 13.8, 4.4 Hz, 1H), 3.40 (dd, *J* = 8.9, 4.3 Hz, 1H), 3.59 (d, *J* = 13.3 Hz, 1H), 3.70 (d, *J* = 13.3 Hz, 1H), 3.74 (s, 3H), 4.62–4.72 (m, 1H), 5.87 (dd, *J* = 15.7, 1.4 Hz, 1H), 6.84 (dd, *J* = 15.7, 5.6 Hz, 1H), 7.07 (d, *J* = 7.6 Hz, 2H), 7.10–7.34 (m, 14H), 7.39 (d, *J* = 8.9 Hz, 1H). ^13^C NMR (126 MHz) in CDCl_3_, *δ* = 32.11, 36.13, 38.99, 49.52, 51.78, 52.89, 63.05, 121.06, 126.30, 127.14, 127.47, 128.08, 128.45, 128.69, 128.74, 128.95, 129.30, 137.24, 139.07, 141.00, 147.78, 166.67, 173.13. Elemental analysis calcd for C_29_H_32_N_2_O_3_: C, 76.29; H, 7.06; N, 6.14; found: C, 75.97; H, 7.11; N, 6.30.

*Methyl (S,E)-4-((S)-2-((4-nitrobenzyl)amino)-3-phenylpropanamido)-6-phenylhex-2-enoate* (**SPR11**) Column chromatography in light petroleum/EtOAc (2:3). Consistency = yellow powder. *R*_f_ = 0.46 (light petroleum/EtOAc, 2:3). Yield = 54%. ^1^H NMR (500 MHz) in CDCl_3_, *δ* = 1.80–1.90 (m, 1H), 1.90–1.99 (m, 1H), 2.63 (t, *J* = 7.8 Hz, 2H), 2.75 (dd, *J* = 13.5, 8.9 Hz, 1H), 3.20 (dd, *J* = 13.8, 4.1 Hz, 1H), 3.30 (dd, *J* = 8.9, 4.3 Hz, 1H), 3.68 (d, *J* = 14.5 Hz, 1H), 3.74 (s, 3H), 3.78 (d, *J* = 13.8 Hz, 1H), 4.59–4.77 (m, 1H), 5.78 (d, *J* = 15.9 Hz, 1H), 6.83 (dd, *J* = 16.1, 5.6 Hz, 1H), 7.05–7.22 (m, 7H), 7.24–7.36 (m, 4H), 8.09 (d, *J* = 13.8 Hz, 2H). ^13^C NMR (126 MHz) in CDCl3, *δ* = 32.12, 36.05, 39.17, 39.41, 49.61, 51.87, 63.22, 121.09, 123.93, 126.38, 127.39, 128.44, 128.62, 128.75, 129.10, 129.27, 137.03, 140.90, 146.53, 147.47, 147.55, 166.54, 172.61. Elemental analysis calcd for C_29_H_31_N_3_O_5_: C, 69.44; H, 6.23; N, 8.38; found: C, 68.59; H, 5.86; N, 8.57.

*Methyl (S,E)-4-((S)-2-((4-methoxybenzyl)amino)-3-phenylpropanamido)-6-phenylhex-2-enoate* (**SPR12**) Column chromatography in light petroleum/EtOAc (3:2). Consistency = pale yellow powder. *R*_f_ = 0.40 (light petroleum/EtOAc, 3:2). Yield = 47%. ^1^H NMR (500 MHz) in CDCl_3_, *δ* = 1.79–1.88 (m, 1H), 1.89–1.97 (m, 1H), 2.61 (t, *J* = 7.9 Hz, 2H), 2.77 (dd, *J* = 13.9, 8.9 Hz, 1H), 3.19 (dd, *J* = 13.8, 4.4 Hz, 1H), 3.38 (dd, *J* = 8.9, 4.3 Hz, 1H), 3.52 (d, *J* = 13.1 Hz, 1H), 3.63 (d, *J* = 13.1 Hz, 1H), 3.74 (s, 3H), 3.78 (s, 3H), 5.87 (dd, *J* = 15.7, 1.6 Hz, 1H), 6.80 (d, *J* = 8.6 Hz, 2H), 6.85 (dd, *J* = 15.7, 5.6 Hz, 1H), 6.98 (d, *J* = 8.5 Hz, 2H), 7.09–7.21 (m, 6H), 7.22–7.32 (m, 6H), 7.42 (d, *J* = 8.9 Hz, 1H). ^13^C NMR (126 MHz) in CDCl3, *δ* = 32.10, 36.13, 38.93, 49.48, 51.77, 52.28, 55.37, 62.84, 114.09, 120.99, 126.27, 127.09, 128.44, 128.67, 128.90, 129.15, 129.29, 131.18, 137.28, 140.99, 147.82, 158.98, 166.67, 173.19. Elemental analysis calcd for C_30_H_34_N_2_O_4_: C, 74.05; H, 7.04; N, 5.76; found: C, 73.88; H, 7.27; N, 5.86.

*Methyl (S,E)-4-((S)-2-((2-fluorobenzyl)amino)-3-phenylpropanamido)-6-phenylhex-2-enoate* (**SPR13**) Column chromatography in light petroleum/EtOAc (7:3). Consistency = pale yellow powder. *R*_f_ = 0.40 (light petroleum/EtOAc, 7:3). Yield = 50%. ^1^H NMR (500 MHz) in CDCl_3_, *δ* = ^1^H NMR (500 MHz) in CDCl_3_, *δ* = 1.80- 2.00 (m, 2H), 2.62 (t, *J* = 7.8 Hz, 2H), 2.72 (dd, *J* = 14.1, 9.5 Hz, 1H), 3.21 (dd, *J* = 13.5, 3.5 Hz, 1H), 3.38 (dd, *J* = 9.3, 3.9 Hz, 1H), 3.55 (d, *J* = 13.1 Hz, 1H), 3.74 (s, 3H), 3.77 (d, *J* = 13.1 Hz, 1H), 4.62–4.74 (m, 1H), 5.91 (dd, *J* = 15.7, 1.4 Hz, 1H), 6.87 (dd, *J* = 15.7, 5.8 Hz, 1H), 6.93–7.09 (m, 3H), 7.10–7.33 (m, 12H), 7.55 (d, *J* = 8.8 Hz, 1H). ^13^C NMR (126 MHz) in CDCl_3_, *δ* = 32.12, 36.08, 39.15, 47.22, 49.58, 51.76, 63.03, 115.67 (d, *J* = 21.7 Hz), 121.12, 124.28 (d, *J* = 3.5 Hz), 125.96 (d, *J* = 14.8 Hz), 126.27, 127.10, 128.46, 128.67, 128.92, 129.16, 129.44 (d, *J* = 8.3 Hz), 130.60 (d, *J* = 4.6 Hz), 137.17, 141.03, 147.76, 161.43 (d, *J* = 246.1 Hz), 166.74, 173.00. Elemental analysis calcd for C_29_H_31_FN_2_O_3_: C, 73.40; H, 6.58; N, 5.90; found: C, 73.23; H, 6.82; N, 6.17.

*Methyl (S,E)-4-((S)-2-((3-fluorobenzyl)amino)-3-phenylpropanamido)-6-phenylhex-2-enoate* (**SPR14**) Column chromatography in light petroleum/EtOAc (7:3). Consistency = pale yellow powder. *R*_f_ = 0.40 (light petroleum/EtOAc, 7:3). Yield = 41%. ^1^H NMR (500 MHz) in CDCl_3_, *δ* = 1.81–1.99 (m, 2H), 2.62 (t, *J* = 7.9 Hz, 2H), 2.75 (dd, *J* = 13.8, 9.1 Hz, 1H), 3.20 (dd, *J* = 13.8, 4.1 Hz, 1H), 3.36 (dd, *J* = 8.7, 4.0 Hz, 1H), 3.57 (d, *J* = 13.8 Hz, 1H), 3.69 (d, *J* = 13.8 Hz, 1H), 3.74 (s, 3H), 4.63–4.73 (m, 1H), 5.84 (d, *J* = 15.7 Hz, 1H), 6.74 (d, *J* = 8.7 Hz, 1H), 6.84 (dd, *J* = 15.0, 6.3 Hz, 1H), 6.82–6.88 (m, 1H), 6.92 (t, *J* = 8.4 Hz, 1H), 7.09–7.36 (m, 13H). ^13^C NMR (126 MHz) in CDCl_3_, *δ* = 32.12, 36.08, 39.10, 49.55, 51.78, 52.23, 63.07, 114.36 (d, *J* = 21.2 Hz), 114.79 (d, *J* = 21.4 Hz), 121.11, 123.59 (d, *J* = 2.8 Hz), 126.32, 127.26, 128.45, 128.71, 129.00, 129.24, 130.20 (d, *J* = 8.2 Hz), 137.12, 140.96, 141.64 (d, *J* = 6.7 Hz), 147.64, 163.05 (d, *J* = 246.4 Hz), 166.62, 172.92. Elemental analysis calcd for C_29_H_31_FN_2_O_3_: C, 73.40; H, 6.58; N, 5.90; found: C, 73.22; H, 6.84; N, 6.19.

*Methyl (S,E)-4-((S)-2-((4-fluorobenzyl)amino)-3-phenylpropanamido)-6-phenylhex-2-enoate* (**SPR15**) Column chromatography in light petroleum/EtOAc (7:3). Consistency = pale yellow powder. *R*_f_ = 0.41 (light petroleum/EtOAc, 7:3). Yield = 51%. ^1^H NMR (500 MHz) in CDCl_3_, *δ* = 1.79–1.89 (m, 1H), 1.90–1.99 (m, 1H), 2.62 (t, J = 7.8 Hz, 2H), 2.75 (dd, J = 13.8, 9.0 Hz, 1H), 3.20 (dd, J = 13.8, 4.1 Hz, 1H), 3.35 (dd, J = 9.0, 4.3 Hz, 1H), 3.54 (d, J = 13.4 Hz, 1H), 3.65 (d, J = 13.4 Hz, 1H), 3.74 (s, 3H), 4.62–4.73 (m, 1H), 5.83 (dd, J = 15.7, 1.5 Hz, 1H), 6.84 (dd, J = 15.7, 5.6 Hz, 1H), 6.94 (t, J = 8.7 Hz, 2H), 6.98–7.04 (m, 2H), 7.08–7.22 (m, 6H), 7.23–7.34 (m, 6H). ^13^C NMR (126 MHz) in CDCl_3_, *δ* = 32.11, 36.12, 39.03, 49.51, 51.81, 52.04, 62.91, 115.54 (d, *J* = 21.4 Hz), 121.04, 126.32, 127.19, 128.44, 128.71, 128.97, 129.27, 129.65 (d, *J* = 8.0 Hz), 134.78 (d, *J* = 3.3 Hz), 137.20, 140.97, 147.72, 162.17 (d, *J* = 245.5 Hz), 166.63, 173.00. Elemental analysis calcd for C_29_H_31_FN_2_O_3_: C, 73.40; H, 6.58; N, 5.90; found: C, 73.19; H, 6.80; N, 6.16.

*Methyl (S,E)-4-((S)-2-((2,4-difluorobenzyl)amino)-3-phenylpropanamido)-6-phenylhex-2-enoate* (**SPR16**) Column chromatography in light petroleum/EtOAc (7:3). *R*_f_ = 0.39 (light petroleum/EtOAc, 7:3). Consistency: pale yellow powder. Yield = 60%. ^1^H NMR (500 MHz) in CDCl_3_, *δ* = 1.82–1.99 (m, 2H), 2.63 (t, *J* = 7.9 Hz, 2H), 2.68 (dd, *J* = 13.9, 9.5 Hz, 1H), 3.21 (dd, *J* = 13.9, 4.1 Hz, 1H), 3.34 (dd, *J* = 9.4, 4.1 Hz, 1H), 3.50 (d, *J* = 13.4 Hz, 1H), 3.72 (d, *J* = 14.4 Hz, 1H), 3.74 (s, 3H), 4.62–4.73 (m, 1H), 5.88 (dd, *J* = 15.7, 1.5 Hz, 1H), 6.67–6.80 (m, 2H), 6.87 (dd, *J* = 15.7, 5.8 Hz, 1H), 6.95–7.03 (m, 1H), 7.10–7.17 (m, 4H), 7.17–7.22 (m, 1H), 7.23–7.31 (m, 5H), 7.47 (d, *J* = 8.9 Hz, 1H). ^13^C NMR (126 MHz) in CDCl_3_, *δ* = 32.11, 36.08, 39.17, 46.62, 49.56, 51.79, 62.93, 104.22 (t, *J* = 25.6 Hz), 111.24 (dd, *J* = 21.0, 3.6 Hz), 121.10, 121.93 (dd, *J* = 15.1, 3.9 Hz), 126.29, 127.15, 128.44, 128.68, 128.94, 129.12, 131.25 (dd, *J* = 9.6, 6.2 Hz), 137.12, 140.99, 147.69, 160.96 (dd, *J* = 145.4, 12.1 Hz), 162.94 (dd, *J* = 145.4, 12.1 Hz), 166.69, 172.87. Elemental analysis calcd for C_29_H_30_F_2_N_2_O_3_: C, 70.71; H, 6.14; N, 5.69; found: C, 70.58; H, 5.94; N, 5.41.

*Methyl (S,E)-4-((S)-2-((2,5-difluorobenzyl)amino)-3-phenylpropanamido)-6-phenylhex-2-enoate* (**SPR17**) Column chromatography in light petroleum/EtOAc (7:3). *R*_f_ = 0.39 (light petroleum/EtOAc, 7:3). Consistency: pale yellow powder. Yield = 64%. ^1^H NMR (500 MHz) in CDCl_3_, *δ* = 1.83–2.00 (m, 2H), 2.63 (t, *J* = 7.8 Hz, 2H), 2.70 (dd, *J* = 13.8, 9.4 Hz, 1H), 3.22 (dd, *J* = 13.8, 4.0 Hz, 1H), 3.35 (dd, *J* = 9.3, 4.0 Hz, 1H), 3.54 (d, *J* = 13.7 Hz, 1H), 3.69–3.76 (m, 1H), 3.74 (s, 3H), 5.88 (d, *J* = 15.7 Hz, 1H), 6.71–6.77 (m, 1H), 6.83–6.86 (m, 3H), 6.86 (dd, *J* = 15.7, 5.7 Hz, 1H), 7.11–7-22 (m, 5H), 7.22–7.33 (m, 6H), 7.40 (d, *J* = 8.8 Hz, 1H). ^13^C NMR (126 MHz) in CDCl_3_, *δ* = 32.10, 36.02, 39.19, 46.72, 49.59, 51.75, 63.11, 115.50 (dd, *J* = 23.9, 8.7 Hz), 116.56 (dd, *J* = 8.4, 6.9 Hz), 116, 75 (dd, *J* = 9.0, 5.4 Hz), 121.13, 126.28, 127.21, 127.62 (dd, *J* = 17.6, 7.2 Hz), 128.44, 128.67, 128.97, 129.11, 137.03, 140.98, 147.61, 156.92 (dd, *J* = 177.6, 2.1 Hz), 158.85 (dd, *J* = 178.8, 2.2 Hz), 166.65, 172.75. Elemental analysis calcd for C_29_H_30_F_2_N_2_O_3_: C, 70.71; H, 6.14; N, 5.69; found: C, 70.56; H, 5.93; N, 5.40.

*Methyl (S,E)-4-((S)-2-((2,6-difluorobenzyl)amino)-3-phenylpropanamido)-6-phenylhex-2-enoate* (**SPR18**) Column chromatography in light petroleum/EtOAc (7:3). *R*_f_ = 0.39 (light petroleum/EtOAc, 7:3). Consistency: pale yellow powder. Yield = 67%. ^1^H NMR (500 MHz) in CDCl_3_, *δ* = 1.81–2.02 (m, 2H), 2.62 (t, *J* = 7.9 Hz, 2H), 2.70 (dd, *J* = 14.0, 9.4 Hz, 1H), 3.21 (dd, *J* = 13.9, 3.6 Hz, 1H), 3.37 (dd, *J* = 9.4, 3.9 Hz, 1H), 3.69 (d, *J* = 13.1 Hz, 1H), 3.75 (s, 3H), 3.72–3.77 (m, 1H), 4.61–4.77 (m, 1H), 5.93 (dd, *J* = 15.7, 1.5 Hz, 1H), 6.77–6.85 (m, 2H), 6.88 (dd, *J* = 15.7, 5.8 Hz, 1H), 7.08–7.32 (m, 12H), 7.54 (d, *J* = 8.7 Hz, 1H). ^13^C NMR (126 MHz) in CDCl_3_, *δ* = 32.10, 36.04, 39.08, 39.97, 49.59, 51.74, 62.95, 111.43 (dd, *J* = 20.4, 5.7 Hz), 114.67 (t, *J* = 19.7 Hz), 121.14, 126.24, 127.06, 128.44, 128.64, 128.88, 129.04, 129.41 (t, *J* = 10.4 Hz), 137.01, 141.03, 147.68, 161.61 (dd, *J* = 248.0, 8.1 Hz), 166.77, 172.74. Elemental analysis calcd for C_29_H_30_F_2_N_2_O_3_: C, 70.71; H, 6.14; N, 5.69; found: C, 70.55; H, 5.91; N, 5.38.

*Methyl (S,E)-4-((S)-2-((4-fluorobenzyl)amino)-3-phenylpropanamido)-6-phenylhex-2-enoate* (**SPR19**) Column chromatography in light petroleum/EtOAc (3:2). *R*_f_ = 0.34 (light petroleum/EtOAc, 3:2). Consistency: pale yellow powder. Yield = 52%. ^1^H NMR (500 MHz) in CDCl_3_, *δ* = = 1.83–1.91 (m, 1H), 1.91–2.00 (m, 1H), 2.63 (t, *J* = 7.8 Hz, 2H), 2.75 (dd, *J* = 13.8, 9.0 Hz, 1H), 3.20 (dd, *J* = 13.8, 4.3 Hz, 1H), 3.33 (dd, *J* = 9.1, 4.4 Hz, 1H), 3.62 (d, *J* = 14.0 Hz, 1H), 3.74 (s, 3H), 3.72–3.78 (m, 1H), 4.62–4.76 (m, 1H), 5.84 (dd, *J* = 15.6, 1.3 Hz, 1H), 6.86 (dd, *J* = 15.7, 5.6 Hz, 1H), 7.11–7.23 (m, 7H), 7.23–7.32 (m, 6H), 7.51 (d, *J* = 7.9 Hz, 2H). ^13^C NMR (126 MHz) in CDCl_3_, *δ* = 32.11, 36.09, 39.05, 49.57, 51.80, 52.11, 62.99, 121.06, 124.20 (q, *J* = 271.2 Hz), 125.64 (q, *J* = 3.7 Hz), 126.34, 127.27, 128.23, 128.23 (q, *J* = 35.8 Hz), 128.43, 128.72, 129.02, 129.24, 129.71 (q, *J* = 32.4 Hz), 137.08, 140.93, 143.02, 147.66, 166.60, 172.81. Elemental analysis calcd for C_30_H_31_F_3_N_2_O_3_: C, 68.69; H, 5.96; N, 5.34; found: C, 68.42; H, 6.11; N, 5.63.

*(S,E)-methyl 4-((S)-2-(4-fluorobenzamido)-3-phenylpropanamido)-6-phenylhex-2-enoate* (**SPR34**) Column chromatography in light petroleum/EtOAc (3:2). *R*_f_ = 0.40 (light petroleum/EtOAc, 3:2). Consistency: white powder. Yield = 51%. ^1^H NMR (500 MHz) in CDCl_3_, *δ* = 1.67–1.88 (m, 2H), 2.48–2.60 (m, 2H), 3.12–3.21 (m, 1H), 3.22–3.31 (m, 1H), 3.78 (s, 3H), 4.52–4.62 (m, 1H), 4.89–4.98 (m, 1H), 5.73 (d, *J* = 15.7 Hz, 1H), 6.52 (d, *J* = 7.9 Hz, 1H), 6.70 (dd, *J* = 15.6, 5.7 Hz, 1H), 6.98–7.13 (m, 5H), 7.14–7.36 (m, 8H), 7.75 (d, *J* = 5.1 Hz, 2H). ^13^C NMR (126 MHz) in CDCl_3_, *δ* = 31.93, 35.90, 38.80, 50.11, 51.79, 55.44, 121.35, 126.30, 127.41, 128.37, 128.62, 128.95, 129.48, 136.42, 140.70, 147.02, 166.47, 166.62, 170.63. Elemental analysis calcd for C_29_H_29_FN_2_O_4_: C, 71.29; H, 5.98; N, 5.73; found: C, 71.36; H, 6.28; N, 5.69.

### 3.2. Biological Evaluation

#### 3.2.1. Enzyme Assays towards Rhodesain

Rhodesain was recombinantly expressed as previously described and Cbz-Phe-Arg-AMC (10 µM) was used as the fluorogenic substrate [75]. The rhodesain-mediated hydrolysis released the fluorescent portion AMC, and fluorescence was measured at room temperature over a period of 30 min, using an Infinite 200 PRO microplate reader, and excitation (380 nm) and emission (460 nM) filters were employed. The assay buffer was composed of 50 mM sodium acetate, 5 mM EDTA, 200 mM NaCl, and 0.005% Brij 35 with a pH equal to 5.5. Similarly, the enzyme buffer contained the same chemicals as the assay, with the exception of Brij, which was replaced by 1,4-dithiothreitol (DTT). All the SPR derivatives were solubilized in DMSO, which was also used as the negative control. E-64 protease inhibitor [76] was employed as the positive control. All the assayed compounds showed a high percentage of inhibition at the screening concentration (>80% at 100 µM), and for this reason **SPR10**–**SPR19** and **SPR34** were appropriately diluted and *K*_i_, *k*_inact_, and *k*_2nd_ values were determined [49,54]. Each independent assay was performed twice and in duplicate, using 96 well-plates in a total volume of 200 µM. The first order inactivation rate constants (*k*_obs_) were determined by means of nonlinear regression analysis of the progress curves (fluorescence (F) vs. time), and the equation F = A (1 − exp(−*k*_obs_t)) + B was used. The obtained *k*_obs_ values were fitted vs. the concentrations of tested inhibitors, and the equation *k*_obs_ = *k*_inact_ [I]/(*K*_iapp_ + [I]) provided the *K*_iapp_ and *k*_inact_ values. *K*_i_ values were obtained using the equation *K*_i_ = *K*_iapp_/(1 + [S]/K_m_), where *K*_m_ is equal to 0.9 µM [77]. Grafit software was used for the nonlinear regression analysis to calculate *k*_inact_ and *K*_i_. The second-order rate constants *k*_2nd_ were directly determined as *k*_2nd_ = *k*_inact_/*K*_i_.

#### 3.2.2. Enzyme Assays towards hCatL

Similarly to rhodesain, the activity towards hCatL was evaluated as reported in the literature [78]. The biological evaluation towards hCatL was performed using the same buffers, reader, fluorogenic substrate, and positive and negative controls as described above for rhodesain, and the preliminary screening was performed at 100 µM. In this case too, each independent assay was performed twice and in duplicate. Differently from rhodesain, *K*_m_ is equal to 6.5 µM [77].

#### 3.2.3. Antitrypanosomal Activity

The parasites used in this study were culture-adapted *Trypanosoma brucei brucei* of the cell line 449, descendants of the Lister strain 427 [79,80]. The cells were grown in HMI-9 medium supplemented with 10% FCS, 50 U/mL penicillin, 50 µg/mL streptomycin, and 0.2 µg/mL phleomycin, at 37 °C and 5% CO_2_. EC_50_ values for the antitrypanosomal activity were determined using the ATPlite assay as previously described [34,49,81]. Briefly, 10 mM compound stocks in DMSO were diluted to 333 µM in HMI-9 medium. This mixture was used to prepare nine further consecutive 1:2 dilutions in HMI-9 medium. Subsequently, 10 µL of diluted compound mixtures were added to 90 µL of media containing 2500 cell/mL in wells of a 96-well plate, in two separate triplicates. The highest DMSO concentration reached in compound plus cells mixture was 0.3%, and parasites treated with DMSO only were used as control. The cells were incubated for 24 h at 37 °C. Subsequently, 50 µL ATPlite 1 step solution (PerkinElmer) was added to each well and luminescence was measured using a CLARIOstar Plus plate reader (BMG Labtech).

#### 3.2.4. Cytotoxicity Assay

HEK293 cells were cultured in high-glucose DMEM medium with L-glutamine, supplemented with 10% FCS, 20 U/mL penicillin, and 20 µg/mL streptomycin, at 37 °C and 5% CO_2_. For the cytotoxicity assay, the cells were seeded in 400 µL at 60,000 cells per well of poly-lysine-coated 48-well plates, and incubated for 24 h at 37 °C. Compound stocks were diluted in DMSO in nine consecutive 1:2 steps, resulting in a set of ten dilutions with concentrations from 7 mM to 13.57 µM. From each dilution, 4 µL were added to the cells in the 48-well plates, in two separate triplicates. Addition of only DMSO served as control. The plates containing cells plus compound dilutions were incubated for 21 h at 37 °C. Subsequently, the medium in each well was exchanged to a medium with resazurin (15 µg/mL), resulting in a final concentration of 6 µg resazurin/well. The plates were incubated for 3 h at 37 °C. Finally, 100 µL of the solution from each well of the 48-well plate was transferred into a 96-well plate, and fluorescence was measured (excitation: 540-14, emission: 590-20) using a CLARIOstar Plus plate reader (BMG Labtech).

### 3.3. Molecular Modeling Methods

Using the Maestro suite [82], we modeled the ligands **SPR16** and **SPR18**. The crystal structures of rhodesain and the human cathepsin L were downloaded from the Protein Data Bank (PDB IDs 2P86 [67] and 3OF9 [69], respectively) and prepared for docking using the Protein Preparation Wizard tool in Maestro. Then, covalent docking simulations were performed using the CovDock [68] module available in Glide. In this protocol, C25 was specified as a reactive residue in the receptor, Michael addition as reaction type, and α, β-unsaturated carbonyl group as ligand functional group represented by a SMARTS pattern [C,c]=[C,c]-[C,c,S,s]=[O] were selected. The docking score was calculated by preferring the conformation of the pose that has the lowest binding free energy. All the images were rendered using the UCSF Chimera Molecular Modeling Software (San Francisco, CA, USA) [83].

## 4. Conclusions

In this paper, we reported the synthesis, biological evaluation, and molecular docking of a small panel of Michael acceptors bearing a reduced amide bond between the P2 and P3 site and a vinyl methyl ester warhead. The novel pseudopeptides showed micromolar and sub-micromolar binding affinity against rhodesain, coupled with a remarkable selectivity towards the target. **SPR16** and **SPR18** exhibited single-digit EC_50_ values against *T. b. brucei*. The discrepancy between the enzyme inhibition and cell-based activities of the compounds reported here with respect to the lead compound suggests additional antitrypanosomal effects. The tested pseudopeptides were able to inhibit protozoa growth by a multi-target action against targets other than rhodesain by accumulation in the acidic protozoan compartments due to the different chemical properties. To our knowledge, the molecules described in this paper are the first pseudopeptides carrying a reduced amide bond and a protonatable secondary amine along the peptide backbone endowed with antitrypanosomal activity. With respect to the pseudopeptides **SPR16** and **SPR18**, the lower antitrypanosomal effect shown by the peptide inhibitor **SPR34** emphasizes the crucial role played by the reduced bond for the activity towards the protozoa. Starting from all these findings, further SAR studies with the purpose to develop reduced amide bond-containing molecules as potential anti-HAT agents should be carried out.

## Data Availability

Not applicable.

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
