# Peer review of "Development of Reduced Peptide Bond Pseudopeptide Michael Acceptors for the Treatment of Human African Trypanosomiasis"

_molecules, 2022, doi:10.3390/molecules27123765_

Round 1

Reviewer 1 Report

Previti et al in the manuscript describe a narrow P3 SAR for HAT and the reduced peptide character of this inhs. by the removal of a single CO group in P3 from the lead PS-1. The rational supporting such structural modification was that the functional group is not involved in H-Bond based on the docking studies. The deletion of the CO generate a secondary basic amine in P3. That seems to be tolerated based on the biological results. The series of analogs have been prepared exploring only the substituents on the aromatic ring of the benzylamine in P3.

The authors report the biological results of the amine derivatives SPR10-19 and the compounds shows a different degree of activity, although the same compounds are tested against hCatL has counter screening and  they proved to be selective.

A molecular modeling of the most promising compounds is also presented to explain their binding and selectivity.

Comment to the authors:

In order to prove that the deletion of the P3 CO group enhance the activity of the different analogs the author should provide the biological data of the corresponding amides, not only the comparison with PS-1 considering that the comparison with the SPR10 is not in its favor. Are these compounds and data available? They should provide the data to have more reliable comparison with the original series

P3 SAR it is very limited to the aromatic substitution of the P3 side chain. It should be expanded with other substituents ie aliphatic, heteroaromatic etc to explain the role of P3 and the key interaction required for the activity

Are the data reported in table 1 for PS1 refer to the 49 paper or are they generate simultaneously with all the other SPR10-19 data? It would be much better if they have done in parallel in the same experiment

Protozoan activity evaluation:

The authors compare protozoal activity against T.b. brucei of SPR16 and 18 with PS-1, and they declare the reduced shift between the IC50 with respect the Ki for the new analogs. In order to prove that it is true due to the chemical modification the author have to provide at least the T.b. brucei data for SPR10 to have fear comparison with PS-1.

Molecular modeling:

The docking study suggest a series of key interaction including a double H Bond in P2 with G66. The same H bond is describe for PS-1 H that has an amide instead of the basic amine. The author should describe the bond potential (energy distances and the effect of positive charge) between the two groups.

Furthermore, the P3 side chain is declared to be involved potentially with two residues Q159 and F61 that from the picture are on the opposite side of the protein. In particular, F61 seems far away to have any interaction.   A more precise description of the distances and interaction is required.

In order to understand the selectivity versus Cat-L the different AA pattern in S2 and S3 is describe and it is presented as explanation for the selectivity. The hypothesis is not supported for the data on SPR11 and SPR12 that present quite different activity on Rhodesian but similar activity for h CatL. Can the authors have another hypothesis for the selectivity?

Reviewer 2 Report

Manuscript ID molecules-1727719

As compared to other lifestyle/infectious diseases, drug/drug discovery focuses less on neglected Tropical Diseases such as Trypanosomiasis, Leishmaniasis, Leprosy, Filariasis etc. Considering the toxicity/lack of effectiveness of the current treatment available, it remains a challenge to find affordable, safe and effective treatments for Human African Trypanosomiasis. This paper describes the synthesis and evaluation of pseudopeptide Michael acceptors as rhodesain-targeting anti Trypanosomal agents. Synthetic compounds SPR10-SPR19 were initially evaluated for activity against rhodesain with fluorogenic assays and then assessed for potential activity against the host protein hCatL. Docking studies were also conducted to confirm selectivity of the two compounds (SPR16 and SPR18) against rhodesain and cathepsin L.

Overall, the paper is well written, with a detailed explanation on the synthesis of these novel molecules. As the main focus of this paper is on the anti Trypanosomal activity of the novel synthetic molecules, it be would be appreciated if the authors provided some further information regarding this aspect as described below.

COMMENT 1 - RESULTS - A graphic representation of the biological activity of SPR10 and SPR19 against rhodesain, and hCatL: I would appreciate if the authors could summarise the biological activity of SPR10 and SPR19 against rhodesain and hCatL, along with the relevant controls (DMSO and E-64), in a graphic format along with appropriate statistical significance annotations.

COMMENT 2- Graphical Representation for Trypanocidal activity with statistical analysis : I would appreciate it if the authors could provide a graphical representation indicating the trypanocidal activity of each peptide along with appropriate controls (culture medium, solvent, vehicle) and appropriate statistical comparison with these controls .

COMMENT 3: Lines 608/ 611 -Methods  3.2.1 & 3.2.2  &   - Enzyme assays towards rhodesain / hCatL:. Detailed information on the assay, including the source of reagents and controls, number of replicates, etc., would be appreciated.

COMMENT 4. Line 614- Methods 3.2.3. Antitrypanosomal activity: It would be appreciated if the authors can provided detail of the culture conditions/ strain or isolate of Trypansoma brucei used. The in- vitro culture of Trypanosomes is somewhat challenging in comparison with routine bacteria/eukaryotic cells, and readers would benefit from detailed explanation of the culture conditions.

COMMENT 5 :  Cytotoxicity : Whether the cytotoxicity of the newly synthesized compounds SPR10 and SPR19 was evaluated in normal human cell lines such as Vero cells, and compared with the available data on the existing trypanocidal agents.

COMMENT 6 : Utility of SPR10 and SPR19 against other Trypanosome species : If the authors are able to comment on the predicted activity of these compounds against other Trypanosomes, like T. congolense or trypanosomes in animals, like T. evansi and  T. equiperdum, based on conservation of the residues in the CatL/Rhodesain homologous protein sequence in other species, that would be great.

I recommend the manuscript for minor revision .
